# Molecular Epidemiology of HIV-1 in Eastern Europe and Russia

**DOI:** 10.3390/v14102099

**Published:** 2022-09-22

**Authors:** Maarten A. A. van de Klundert, Anastasiia Antonova, Giulia Di Teodoro, Rafael Ceña Diez, Nikoloz Chkhartishvili, Eva Heger, Anna Kuznetsova, Aleksey Lebedev, Aswathy Narayanan, Ekaterina Ozhmegova, Alexander Pronin, Andrey Shemshura, Alexandr Tumanov, Nico Pfeifer, Rolf Kaiser, Francesco Saladini, Maurizio Zazzi, Francesca Incardona, Marina Bobkova, Anders Sönnerborg

**Affiliations:** 1Division of Infectious Diseases, Department of Medicine Huddinge, Karolinska Institutet, 14152 Stockholm, Sweden; 2T-Lymphotropic Viruses Laboratory, Gamaleya Centre of Epidemiology and Microbiology, 123098 Moscow, Russia; 3EuResist Network, 00152 Rome, Italy; 4Department of Computer Control and Management Engineering Antonio Ruberti, Sapienza University of Rome, 00185 Rome, Italy; 5Infectious Diseases, AIDS and Clinical Immunology Research Center (IDACIRC), 0160 Tbilisi, Georgia; 6Institute of Virology, University of Cologne, 50935 Cologne, Germany; 7Moscow Regional Center for Control and Prevention of AIDS and Infectious Diseases, 123098 Moscow, Russia; 8Clinical Center of HIV/AIDS of the Ministry of Health of Krasnodar Region, 350015 Krasnodar, Russia; 9Methods in Medical Informatics, Department of Computer Science, University of Tübingen, 72076 Tübingen, Germany; 10Department of Medical Biotechnology, University of Siena, 53100 Siena, Italy; 11InformaPRO, 00152 Rome, Italy; 12Division of Clinical Microbiology, Department of Laboratory Medicine, Karolinska Institutet, 17177 Stockholm, Sweden

**Keywords:** Eastern Europe, epidemiology, phylogeny, HIV-1 sub-subtype A6, pre-existing drug resistance mutations, unique recombinant forms

## Abstract

The HIV epidemic in Eastern Europe and Russia is large and not well-controlled. To describe the more recent molecular epidemiology of HIV-1, transmitted drug resistance, and the relationship between the epidemics in this region, we sequenced the *protease* and *reverse transcriptase* genes of HIV-1 from 812 people living with HIV from Ukraine (*n* = 191), Georgia (*n* = 201), and Russia (*n* = 420) before the initiation of antiretroviral therapy. In 190 Ukrainian patients, the *integrase* gene sequence was also determined. The most reported route of transmission was heterosexual contact, followed by intravenous drug use, and men having sex with men (MSM). Several pre-existing drug resistance mutations were found against non-nucleoside reverse transcriptase inhibitors (RTIs) (*n* = 103), protease inhibitors (*n* = 11), and nucleoside analogue RTIs (*n* = 12), mostly polymorphic mutations or revertants. In the *integrase* gene, four strains with accessory integrase strand transfer inhibitor mutations were identified. Sub-subtype A6 caused most of the infections (713/812; 87.8%) in all three countries, including in MSM. In contrast to earlier studies, no clear clusters related to the route of transmission were identified, indicating that, within the region, the exchange of viruses among the different risk groups may occur more often than earlier reported.

## 1. Introduction

According to the HIV/AIDS surveillance in Europe 2021 report by the European Centre for Disease Prevention and Control/WHO Regional Office for Europe [1], the HIV epidemic in Western Europe is largely contained. Intensive monitoring of HIV prevalence and spread in risk groups, availability of testing, and antiretroviral medication and monitoring of therapy effectiveness have reduced the infection rate to 3.7 infections per 100.000 population. By contrast, the HIV epidemic in Eastern Europe is less well contained. In 2020, an estimated 84.556 new HIV infections occurred (rate: 32.6 per 100 000). Most of the infections occurred in Russia (59.598 infections, rate: 40.8) and Ukraine (15.658 infections, rate: 37.5). In Georgia, 530 new cases (rate: 13.3) were reported [1]. Notably, the number of registered new cases published by the Federal Russian AIDS centre was even higher; 88.154 new cases (rate 49.1) [2].

Presumably, the HIV epidemic in this region was initiated by two separate introductions of the virus in intravenous drug users (IDUs) in Ukraine in the mid-1990s [3]. The IDU-A strain, formerly referred to as A-Former Soviet Union (FSU-A) and now classified as sub-subtype A6, led to an HIV-1 outbreak in Odessa [4,5,6]; the IDU-B (FSU-B) strain initiated the epidemic in Mykolaiv. IDU-B differs phylogenetically from other widespread subtypes B—Western B or Thai B—and can be reliably distinguished from them [7]. IDU-A then spread to St. Petersburg, and later throughout all Russian regions [3]. IDU-B remained in Ukraine and spread by drug usage and sexual contacts, both hetero- and homosexual. In Kaliningrad, the outbreak of the HIV-1 epidemic was initiated by a recombinant between the IDU-A and -B viruses [8]. In Russia, most MSM were initially infected with subtype B, which was dominant in Europe at that time.

Ukraine has one of the largest epidemics in Europe. Much like the situation in Russia, the epidemic was initially driven by IDUs, but more recently, sexual intercourse has become the predominant mode of transmission. Successful prevention and treatment programs were set up, leading to the stabilisation of the epidemic since 2012 [9].

The Russian national protocol recommends initiation of antiretroviral therapy (ART) to all patients with HIV infection, as soon as possible after diagnosis, regardless of CD4 count and viral load. The preferred first regimen consists of tenofovir disoproxil fumarate (TDF), lamivudine (3TC), and dolutegravir (DTG), or TDF, emtricitabine (FTC) + DTG, and the alternative regimens are based on abacavir (ABC), 3TC, and DTG, or TDF, 3TC, and low dose (400 mg) efavirenz (EFV). In practice, due to the low cost of EFV, this drug is most often prescribed in the first regimen and replaced with DTG in case of toxic side effects. Ukraine and Georgia have adopted the WHO consolidated guidelines on antiretroviral therapy from 2021 [10], recommending starting as soon as possible after diagnosis, consisting of DTG in combination with an NRTI backbone as the preferred first regimen, or, alternatively, low dose EFV in combination with an NRTI backbone.

Differences in the clinical outcome and in the pattern of drug resistance mutations (DRMs) have been reported between HIV-1 subtypes but the impact is still under investigation. The large noninferiority studies, ATLAS, ATLAS-2M, and FLAIR [11,12,13], demonstrated a high efficacy and safety of a once monthly or bimonthly injectable combination of the integrase strand transfer inhibitor (INSTI) cabotegravir (CAB), together with the non-nucleoside reverse transcriptase inhibitor (NNRTI) rilpivirine (RPV), as a switch strategy in subjects under successful treatment. Few confirmed virological failures (CVF) have been observed [13]. However, for the HIV-1 sub-subtype A6, there has been concern about a greater risk of emergent resistance. Thus, the combination of at least two baseline factors among HIV-1 subtype A1 or sub-subtype A6, a body mass index ≥30 kg/m^2^, and RPV resistance-associated mutations, has been reported to be associated with an increased risk of CVF at week 48 [14]. In Russia, Ukraine, and Georgia, CAB + NNRTI is not used as a switch strategy yet.

To gain updated insights in the molecular epidemiology and the prevalence of DRMs to NNRTIs, nucleoside/nucleotide analogue RTIs (NRTIs), protease inhibitors (PIs), and INSTIs, we analysed the *protease* (PR) and *reverse transcriptase* (RT) sequences of 812 HIV-1 genomes isolated from patients from Ukraine, Russia (Moscow and Krasnodar), and Georgia, as well as 190 HIV-1 *integrase* (IN) sequences from Ukraine.

## 2. Materials and Methods

### 2.1. Study Population

We analysed 812 HIV-1 PR and RT sequences from plasma samples donated by people living with HIV (PLWH) before initiation of ART, who were recruited consecutively at HIV clinics in Moscow and Krasnodar (Russia), Kyiv (Ukraine), and Tbilisi (Georgia). HIV-positive status was determined by double repeated enzyme immunoassay and confirmed by Western blot. The demographics, clinical data, and information about the untreated status were collected by physicians from the patient’s medical charts.

From Russia, 364 sequences from the Moscow region and 56 from the Krasnodar region were included. These sequences were produced in the laboratories of the Moscow regional and Gamaleya centres between 6 August 2019 and 22 July 2020. From Georgia, 201 sequences were obtained in 2013–2017 at the Infectious Diseases, AIDS, and Clinical Immunology Research Center (IDACIRC), Tbilisi. From Kyiv, Ukraine, 191 plasma samples of therapy naïve patients were obtained between 18 December 2019 and 20 October 2020. Collection was coordinated by the Public Health Centre of the Ministry of Health of Ukraine (PHC). After collection, plasma samples from Ukraine were frozen and distributed to the Karolinska Institutet, Stockholm, Sweden, the University of Siena, Siena, Italy, or the University of Cologne, Cologne, Germany, for HIV-1 RNA purification and sequence analysis. IN sequences were determined and analysed in the Ukrainian samples only, since the IN sequencing methodology was not used in routine practice at the laboratories in Russia and Georgia and it was not possible to send samples to other laboratories, except from Ukraine.

Ethical approvals were obtained by the local ethical committees in Moscow (Ethics Committee of the Gamaleya Institut, with Protocol n. 16 of 8 February 2019), Kyiv (Institutional Review Board of the Public Health Center MOH of Ukraine, with Protocol n. 19 on 2 December 2019), and Tbilisi (Institutional Review Board of the Infectious Diseases, AIDS, and Clinical Immunology Research Center, with Protocol number # 19-007, on 10 September 2019). Written informed consent was signed by the subjects involved. Material transfer agreement was obtained for the transport and use of plasma samples from Ukraine.

### 2.2. Sequence Determination, Analysis, and Phylogenetics

The pol region from plasma HIV RNA was genotyped by Sanger sequencing by the ViroSeq HIV-1 Genotyping System (Abbott Molecular, Des Plaines, IL, USA) as per manufacturer’s instructions or in-house methods for Ukrainian samples [15]. Sequencing was performed using an automated Genetic Analyzer ABI Prism 3130 (Applied Biosystems, Foster City, CA, USA). The sequences were aligned and quality control was performed using ClustalW multiple sequence alignment function [16] and in BioEdit [17] and in MEGA11 [18]. For the identification and classification of drug resistance mutations, the Stanford HIVdb (version 9.1) was used [19,20].

Phylogenetic trees were generated and visualised using a local instance of the Nextstrain workflow [21]. The local Nextstrain instance was constructed with Python v3.7.5 using the nextstrain augur v 6.1.1 tool suite. The initial phylogenetic tree was created using the ‘augur tree’ command with the ‘method’ parameter set to ‘iqtree’ using IQ-TREE v1.6.12 [22]. Afterwards, the tree was refined using the ‘augur refine’ command with the ‘--dateinference’ option set to ‘marginal’ and the ‘--clock-filter-iqd’ option set to 6. This parameter was set relatively high mainly to allow the inclusion of HIV-1 reference sequences of all subtypes. Because of the high divergence, however, we refrained from dating the trees in the manuscript and instead have the branch lengths represent diversity rather than time. To verify the correctness of the phylogenetic analysis, the resulting tree was compared to a neighbour-joining tree that was constructed using the Maximum Composite Likelihood method in MEGA11 (data not shown) [18,23]. The trees used for the analysis of the genotype B and the CRFs was rooted on a subtype C sequence (C.U52953) for the analysis of all single genotype sequences, and the *integrase* sequences a subtype D sequence (D.K0345) [24] was used as a root by using the ‘--root’ option. We also compared the tree of all single subtype sequences to a time-based tree that was generated using BEAST. When we compared this analysis to the nextstrain analysis, which has divergence-based branch lengths and is rooted on an outgroup, we observed that, although the general topology of the trees was highly similar, there were small differences within subclusters, which is probably at least partially due to the fact that, for the Georgian sequences, no exact date was available (only a range). Thus, the reliability of the estimates in the time three is poor (small ESS). Therefore, we conclude that the traces did not indicate reliable results, and therefore took the approach based on the sequences without the infection times.

Patient gender, country and region, risk group and age, and virus subtype, DRMs, and variants were assigned using the ‘augur traits’ command. Ancestral nucleotide sequences were inferred with the ‘augur ancestral’ command with the ‘--inference’ parameter set to ‘joint’. Protein sequences were translated from the nucleotide sequence with the ‘augur translate’ command using an HIV-1 subtype B sequence (Accession D86068) [25] annotated in-house for the IN, RT, and PR coding regions as a reference. This reference was also used to verify the presence and correct assignment of the DRMs by using the visualisation ‘Color By’ Genotype > gene > position with gene sat to the appropriate gene and position to the DRM position. The local instance was then exported with the augur export V2 and visualized with Auspice version v2.34.1. Numerical data analysis was visualised using Microsoft Excel (Appendix A) and Graphpad Prism.

### 2.3. Determination of the HIV-1 Subtype

Preliminary determination of the HIV-1 subtypes was carried out using online reference programs: HIV-1 COMET [26], the HIVdb sequence analysis programs on the Stanford University website [20,27,28], and REGA Tool for Determining HIV-1 Subtypes (V3) [29]. With an unambiguous interpretation of the subtype with a percentage of identity from 97 to 100% in all three programs, the subtype was considered reliably determined. With ambiguous results and/or an identity of less than 97%, the studied nucleotide sequences were subjected to additional analysis to identify recombinant viruses. At first, the search for closely related sequences was carried out using the HIV-BLAST tool [30,31]—when finding several similar sequences with an identity percentage of 97–100%, the subtype of the studied sequences was determined as identical to those found. Samples with an indeterminate subtype were subjected to additional recombination analysis.

Samples with the assumed subtypes CRF02_AG and CRF63_02A6 were further analysed using the Recombination Identification Program (RIP) [32,33], since this program, unlike others, includes reference sequences of these subtypes, as well as through phylogenetic analysis. The subtypes of all sequences were confirmed by phylogenetic analysis. For the identification of the HIV-1 sub-subtype A6, our in-house consensus sequence was used, as described [34].

### 2.4. Recombination Analysis

For the identification and detailed analysis (determination of recombination points) of potential recombinant forms of HIV-1, we used the following tools: jumping profile Hidden Markov Model (jpHMM) [35,36,37], recombination identification program (RIP) [32,33], and recombination detection program (RDP4) [38]. The recombination event in the RDP4 program was considered reliable when it was identified by two or more mathematical models. When working in the RDP program, the reference sequences presented in the Appendix A were used (Appendix A). A graphical image of the genome with recombination points was obtained in the jpHMM program or performed manually according to the results of the RDP4 program using the Recombinant HIV-1 Drawing Tool [39] (Appendix A).

## 3. Results

### 3.1. Demographic Characteristics

The median age of the 812 HIV-1-infected patients was 40 years (range: 6–74 years; 20–74 years, when excluding a single mother-to-child transmission (MTCT) case in Krasnodar; interquartile range: 33–47 years) (Table 1). The percentage of men (62.2%) was higher than that of women. The most likely route of transmission was based on a self-report at diagnosis and a subsequent epidemiological survey done by the health care personnel. The sex and age distribution were similar in the three countries. In all groups, the dominant transmission route was heterosexual contacts (61.3%). Intravenous drug usage (26.1%) and men who have sex with men (MSM: 10.8%) were less common, although the number of MSM we find in Russia is higher than previously reported, at 2.8% by the Russian agency Rospotrebnadzor, which is based on the self-reported route of transmission for all cases in Russia [2]. A relatively low number of patients reported MSM in Georgia (4%) (Table 1).

### 3.2. HIV-1 Subtype Distribution

The subtype of the 812 patients was determined based on the PR and RT sequences. For the nonrecombinant viruses, the subtype, including the sub-subtype A6, was confirmed by phylogenetic analysis using the first 919 bases from the start of the PR open reading frame (ORF) (Figure 1A). In line with previous observations in the region, most of the sequences were sub-subtype A6 (713/812; 87.8%) (Figure 1A,B; Table 2).

The next most common subtype was subtype B (59/812 sequences, 8.4%; Table 2, Figure 1A,B). The phylogenetic relation of the B subtypes identified in our study to archetypical IDU-B (Eastern European) viruses, that initiated the outbreak in Ukraine, and the Western European B-variant was studied by phylogenetic analysis of a 919 bp fragment of the HIV-1 PR/RT gene, using Western B, Thai B, and IDU-B sequences as references (Figure 1C). In contrast to earlier observations [5], viruses from the different regions included in our study could not be distinguished by phylogeny.

A cluster of infections with subtype G among infected persons (*n* = 6) in the Moscow region was identified, as well as a single case of infection with subtype D in Ukraine (Figure 1A,B; Table 2).

### 3.3. Recombinant Viruses

Possible recombinant strains were identified by the analysis of all sequences that were not unambiguously identified as a single subtype or sub-subtype by the COMET HIV-1 program, HIVdb Program for Sequence Analysis, and REGA Subtyping Tool HIV-1 (Appendix A). A first selection of possible recombinant strains was made by using HIV BLAST. Thereafter, the jpHMM and RDP4 programs were used to verify the results (Appendix A). Finally, the results were confirmed by phylogenetic analysis (Figure 1D).

Two main types of circulating recombinant forms (CRF) have been previously identified in Russia. The CRF02_AG, which originated from Cameroon and came to Russia through Uzbekistan, and the CRF63_02A6, which is the result of a recombination between the Uzbek CRF02_AG and the sub-subtype A6 [40]. We distinguished these two CRFs using the Recombination Identification Program and confirmed the results by phylogenetic analysis (Figure 1D), as the Stanford tool tended to define everything as CRF63_02A6, and REGA and COMET gave vague answers (Appendix A). Based on this analysis, 16 sequences were classified as CRF02_AG and 8 as CRF63_02A6. Furthermore, two infections with the CRF03_AB virus, the regionally CRF that initiated the outbreak in Kaliningrad [8,41], were identified. Except for two sequences from Krasnodar, which did not cluster strongly with the other CRF02_AG sequences (Figure 1D), all CRFs were from Moscow.

Moreover, 14 unique recombinant viruses (URFs) were identified (Appendix A), of which 12 were recombinants between A6 and B strains, and 2 were recombinants between subtype B and the CRF02_AG or CRF63_02A6, respectively. A graphical representation of the URFs, including their breakpoints, is provided (Appendix A). Except for one URF that was identified in two patients from Ukraine, all URFs had unique breakpoints (Appendix A and Appendix A).

### 3.4. Transmission Chains

The majority of the sequences analysed (87%) was of sub-subtype A6 (earlier named FSU-A), which initially spread by IDUs, first in Ukraine and later in other parts of the former Soviet Union. Our data suggest that this subtype is now responsible for most of the infections, not only among IDUs, but also by other routes of transmission (Table 3).

To explore the phylogenetic relation among sub-subtype A6 viruses from the different regions, we generated a phylogenetic tree in which the HIV-1 strains were coloured by region (Figure 2A). Clearly, Ukrainian A6 samples tended to cluster closer to the root of the A6 branch, in line with the likely initiation of the A6 epidemic in this country. Viruses from Georgia formed some distinct clusters, as well as single cases, that were rooted at different locations in the tree, indicating that these viruses in our study were introduced to this region on multiple occasions at different timepoints since the start of the epidemic. It should be noted that these observations may be biased by the fact that the Georgian samples in our study were drawn at earlier time points and during a longer period than the samples from Russian and Ukraine.

The most likely transmission routes were inferred using the Nextstrain workflow (Figure 2B). Briefly, the geographical location of internal nodes in the tree was inferred. The size of the individual lines was calculated based on the number of parent–child branches in the phylogenetic tree where the value of the geographic resolution differs and is therefore a measure for the number of transmissions from one area to another. It should be noted that, as such the quantification is biased by the differences in sample sizes, sampling time and the data depicted represent an ancestral state reconstruction rather than a classical epidemiological investigation of transmission events. The largest transmission route was the one from Ukraine to Moscow. Few transmissions from Krasnodar occurred, in line with the small local clusters in the phylogenetic tree. Despite many transmissions from Ukraine to Krasnodar and Georgia, very few transmissions from these regions to Ukraine took place.

A significantly higher percentage of MSM were infected with subtype B viruses (24%, versus 7% of total cases, *p* < 0.001). Phylogenetic analysis did not reveal a clear clustering of the circulating viruses to reference strains of the phylogenetically distinct B strains predominant in Western Europe, Thai genotype B strains, and/or the B strain that initially spread in Ukraine (Eastern European B or B-FSU) (Figure 1C). This is in contrast to previous findings, where the western B-variant was predominantly found in MSM in Russia, and the IDU-B variant was restricted to IDU and heterosexually (HTX) related infections in Ukraine. If anything, the subtype B strains identified in our study could be described as being “remote” from the archetypical strains.

### 3.5. Drug Resistance Mutations

The study participants were included in the study before receiving any ART. The sequences were analysed using the Stanford University HIV Drug Resistance Database to identify pre-existing pDRMs (Appendix A). By far, the most commonly observed mutations were NNRTI mutations, which were found in 106 (12.7%) of the isolates. These mutations were scattered among the sequences analysed, with no large transmission clusters of strains with DRMs (Appendix A). The most commonly observed nonpolymorphic mutations were the V106I (*n* = 8; 1.0%) and the K103N (*n* = 5; 0.6%). In addition, the V106VI (*n* = 7; 0.9%) and the K103KN (*n* = 3; 0.4%) were identified. V106I is an NNRTI-selected mutation that also occurs in 1% to 2% of viruses from untreated persons, and K103N is a nonpolymorphic mutation selected in patients receiving nevirapine (NVP) and/or EFV [20].

In total, eight different single NRTI mutations were observed (E44D, T69DN, M41L, M41ML, K70KIM, M184MI, T215TI, and K219KR) each in an individual patient. These mutations, when occurring in combinations, are mostly associated with reduced susceptibility to zidovudine (AZT), didanosine (ddI), and stavudine (d4T). Single mutations can reflect the process of reversion of resistant viruses to the wild phenotype. In addition, three phylogenetically unrelated viruses (Appendix A) with multiple DRMs were observed. These patients all carried K65R and M184V, with one patient additionally carrying the Y115F mutation and another one the Y115YF mutation. The K65R mutation reduces susceptibility to TDF, 3TC/FTC, ABC, and ddI. M184V reduces susceptibility to 3TC/FTC. Both mutations reduce viral fitness and are only observed in patients failing combination therapy. Moreover, one patient with the D30N PI mutation and several NRTI mutations was found (Appendix A). Comparable analysis of other DRMs identified indicated there was no phylogenetic relation between the different viruses carrying DRMs.

Several PI mutations were observed. The most common was at position M46 (46I: 3, 46MI: 3, 46L: 2). These mutations are associated with reduced susceptibility to atazanavir (ATV), fosamprenavir (FPV), indinavir (IDV), lopinavir/ritonavir (LPV), and nelfinavir (NFV). Moreover, the nonpolymorphic D30N mutation, which is selected by NFV (*n* = 2), as well as the nonpolymorphic N88NS mutation, which is selected by NFV, ATV, and IDV (*n* = 1), were found. A D30N patient also had several NRTI (D67N, K70E, M184I) and NNRTI (K101E, Y181C, H221HY) DRMs.

We also determined the IN sequence of the Ukrainian patients (*n* = 190). The sequences clustered well by the subtypes previously determined based on the PR and RT genes (Appendix A). No major INSTI DRMs were found.

### 3.6. Polymorphic Mutations

The accessory NRTI A62V mutation, which partially restores the replication defect associated with the K65R mutation, was found exclusively in patients (210/700, 30%) infected with sub-subtype A6, a prevalence which is somewhat lower than previously obtained data for sub-subtype A6 (48–54%) [42]. This may be due to a strong clustering of this variant and therefore presumably a higher prevalence in some regions not included in our study. In our study, the mutation appeared both in a large cluster and, apparently, spontaneously in many individuals (Appendix A). Moreover, the A62AV (transitioning) variants were commonly observed both within and outside the cluster.

The most common mutation was the E138A (*n* = 42; 5.2%). In addition, the E138EA (*n* = 7; 0.9%) were identified. The E138A can be weakly selected by etravirine and RPV. We found it in 5.8% of the patients with an association with sub-subtype A6, which is in line with earlier report in sub-subtype A6, where the frequency was 4–8% depending on the region [38]. It has recently been reported that this singleton mutation, before starting treatment with RPV, may not affect the efficacy of the ART [43]. In the *protease* ORF, the L33F (0.7%) polymorphism was found, which is similar to the frequency observed in other subtypes.

To identify pDRMs in the IN protein, we sequenced the IN gene in 190 Ukrainian patients. Phylogenetic analysis revealed that all sequences clustered with the genotype previously identified by analysis of the RT and PR genes, except for one virus that was found to be a URF between B and A6, with the breakpoint between the sequenced part of the reverse transcriptase and the IN sequences (Appendix A). The INSTI accessory mutation E157Q occurred three times in a cluster of patients infected with subtype B (Appendix A), and the T97TA occurred once in a patient infected with sub-subtype A6. These patients did not have any other pDRMs. The L74I [11,12,13] was found in 164 of 176 (93.3%) A6 isolates (Appendix A), which is in line with previous findings in the sub-subtype A6 [42,44], and in the two B/A6 URF (Appendix A) from Ukraine.

## 4. Discussion

We analysed the HIV-1 sequences of 812 persons from Russia, Ukraine, and Georgia before their first ART regimen. Our results suggest that the vast majority (87.8%) of the cases in this region are still caused by the sub-subtype A6, which initiated the epidemic amongst IDUs in Ukraine in the mid-1990s, and later in other parts of the former Soviet Union. However, it should be noted that a change in the sub-subtype A6 distribution seems to have occurred during these years, since A6 is now dominating also among MSM. In addition, almost all URFs consisted of A6 and B sequences. Together with our finding that the two variants of subtype B were no longer concentrated in different transmission groups, the data suggest an increased spread and intermingling of virus strains between the groups.

The most self-reported route of infection in our study was heterosexual contact (61%), followed by IDU (26%) and MSM (11%). These numbers are comparable to those gathered by the Russian agency Rospotrebnadzor, which estimated in 2020 that of the total number of PLWH in Russia (1,492,998), 64.9% were infected by heterosexual contact, 31.1% by IDU, and 2.8% by MSM [2].

In our study, the most likely transmission routes were inferred from the phylogenetic analysis. Although this method is not as accurate as classic epidemiological investigation and is biased by sample size and differences in the observed periods, our conclusions are in line with historical observations. In line with earlier studies, our molecular epidemiology analysis gives support for the view that the A6 epidemic started in Ukraine and spread further in Eastern Europe [3,4,5,6]. The largest was the one from Ukraine to Moscow. Few transmissions from Krasnodar occurred. Despite many transmissions from Ukraine to Krasnodar and Georgia, few transmissions from these regions to Ukraine seemed to have taken place in our cohorts. If we assume that risk behaviour implies grossly equal chances of transmission in both directions, it seems unlikely that this pattern is observed because only transmissions from Ukraine took place. Rather, they may represent transmissions early in the epidemic, when all A6 viruses were of the typical Ukrainian strain, or transmissions from Russia by viruses that were introduced there more recently from Ukraine. Moreover, they may represent transmission from a location that was not included in this study.

The next most common subtype was subtype B (59/812 sequences). In the past, the archetypical IDU-B virus for Eastern Europe dominated in Ukraine, and the Western European B-variant in Russia. In contrast to earlier observations in the region, B-strains from Ukraine and Russia could not be distinguished by phylogeny in our study [7]. The analysis indicates that the two subtype B variants are no longer concentrated in different transmission groups and that subtype B has evolved away from the two archetypical B-strains, which appeared in the beginning of the HIV-1 epidemic in Ukraine and Russia.

In total, 40 recombinant viruses were identified, mostly the previously described CRF02_AG and CRF63_02A6, but also 14 URFs, which almost all (12/14) were between recombinants between A6 and B. The URFs appeared not to be spreading through the population, as only one new URF was found in two different patients from Ukraine. However, the number of involved subtypes, sub-subtypes, and CRFs recombining to create the URFs found in the present study are also in line with the spread of HIV-1 strains among individuals outside closed social networks.

The HIV sequences presented in this study were obtained at diagnosis, before the initiation of ART. Several pre-existing DRMs were identified, most frequently (12.7%) NNRTI DRMs. However, the most common was E138A (5.2%), a polymorphic mutation common in sub-subtype A6, which seems not to affect the response to RPV. Interestingly, some of these mutations were probably in the process of reverting. This indicate that there was a selection pressure to revert making it less likely that the mutations occurred spontaneously. This may be due to either unreported earlier treatment with NNRTIs or transmitted virus with these mutations. Although less common, pre-existing DRMs against NRTIs and PIs were also found. These numbers are higher than those found in Russia between 1998 and 2017 (5.3%) [45]. At that time, the prevalence of M184MI was 0.38%, and K103N was 1.15%. Moreover, the number of PDRs was slightly higher than other recent studies in the region. A study from 2022 estimated that the prevalence of PDR to any drug class was 2.8% in Uzbekistan, 4.2% in Azerbaijan, 4.5% in Russia (7% in Krasnodar), 9.2% in Armenia, 13.9% in Belarus, and 16.7% in Tajikistan [42], although it should be noted that, due to the limited availability of data, these studies were not statistically significant according to WHO recommendations. The only statistically reliable study in the region was carried out in Uzbekistan in 2016, when the PDR was found to be 2.96% [46].

A high prevalence of the L74I mutation (93%) was found in the IN gene of Ukrainian A6 strains, consistent with earlier data [42,44]. A once monthly or bimonthly injectable combination of the INSTI CAB and the NNRTI RPV has resulted in confirmed virological failures. The combination of at least two baseline factors among HIV-1 subtype A1 or sub-subtype A6, a body mass index ≥30 kg/m^2^, and RPV resistance-associated mutations has been reported to be associated with an increased risk of treatment failure [8,9,10,11]. However, not all of the patients who developed treatment failure and resistance had these risk factors.

One of the limitations of this study is that the data came from four different sources with different sampling techniques and different data collection methods for self-reported transmission, as well as at five different laboratories. However, the study was performed within a larger collaborative study, the CARE study, in which we aimed at the harmonization of the approaches. MSM are often stigmatized and, as a consequence, they might have been underreported in our study, and thereby the distribution of subtypes across the different transmission categories may have not been fully correct. The patients were recruited during a similar time period in Ukraine, Moscow, and Krasnodar, but at earlier time points in Tbilisi. This is likely to have contributed to the fact that the Georgian strains were rooted at different locations in the phylogenetic tree, indicating introduction on multiple occasions at different timepoints since the start of the epidemic.

In conclusion, our data suggest that the HIV-1 epidemic in Eastern Europe is still driven by regional variants, mainly the sub-subtype A6. However, the change in sub-subtype A6 distribution with increased prevalence among MSM, together with the evolution from the archetypical B strains, as well as a significant number of CRFs and URFs, mainly A6/B, suggest an increased spread of the virus and intermingling between the risk groups. The prevalence of pretreatment HIV-1 DRMs was relatively low (12.7%), and at least some of these mutations seem to be natural polymorphisms rather than transmitted drug resistance. However, the numbers are higher than those previously reported, indicating that there is still an important need to monitor transmission of DRMs before initiation of ART, and thus continuous surveillance is warranted.

## Figures and Tables

**Figure 1 viruses-14-02099-f001:**
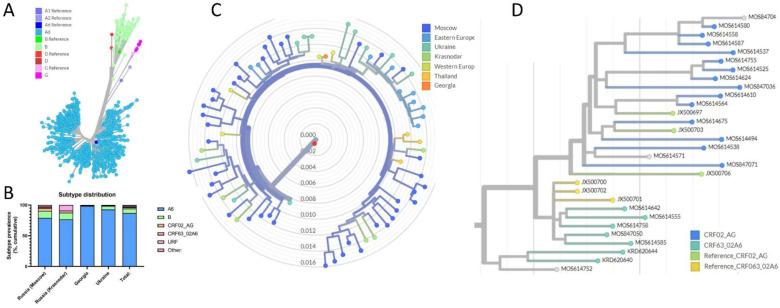
Sequence analysis of the HIV-1 *pol* gene. (**A**): Unrooted phylogenetic analysis. Colours were assigned based on sequence analysis using HIV COMET, the Stanford database tool, and the REGA tool. HIV-1 subtype A1, A2, A, B, D, and G reference sequences were included. (**B**): Distribution of HIV-1 subtypes, sub-subtypes, and recombinant forms in the different countries and regions studied. CRF: Circulating recombinant form. URF: Unique recombinant form. Other: Subtype D (1 case), CRF03_AB (2 cases), and G (6 cases). (**C**): Phylogeny of HIV-1 subtype B viruses identified in this study. The tree was rooted on a genotype C sequence (C.U52953). The numbers indicate the genetic distance (divergence) between the viruses. (**D**): Phylogenetic analysis of CRF02_AG and CRF63_02A6 viruses. The tree was rooted on a subtype C sequence (C.U52953). The samples in grey were conclusively assigned CRF02_AG (MOS84704 and MOS614571) or CRF063_02A6 (MOS614752) based on the phylogenetic analysis.

**Figure 2 viruses-14-02099-f002:**
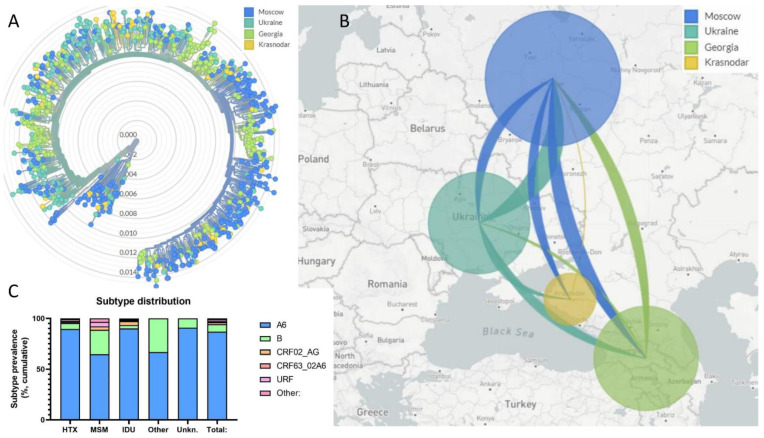
(**A**): Phylogenetic relation between the HIV-1 strains coloured by the locations where the viruses were obtained. The central cluster are the subtype B sequences; the small, Moscow-based cluster between this cluster and the main cluster are the genotype G sequences. The big cluster on the outside are the sub-subtype A6 sequences. The numbers indicate the genetic distance (divergence) between the viruses. The tree was rooted on a subtype D sequence (D.K0345). (**B**): The most likely transmission routes were inferred from the phylogenetic analysis depicted in panel A using the nextstrain workflow. Directionality of transmissions is pictured by the connecting lines, which have the colour of the country from which the transmissions took place. The surface area of the circles represents the number of cases analysed in each region. (**C**): Subtype distribution by transmission route. For transmission route categories see Table 3 legend.

**Table 1 viruses-14-02099-t001:** Characteristics of HIV-1-infected patients from Russia, Georgia, and Ukraine.

	Inclusion		Gender		Route of Transmission				
	Period	Patients	Male	Female	HTX	MSM	IDU	Other	Unkn.
Russia (Mos)	Aug. 2019–Jul. 2020	363	232 (64)	132 (36)	232 (64)	48 (13)	78 (21)	1 (<1)	5 (1)
Russia (Krd)	Aug. 2019–Nov. 2019	56	22 (59)	23 (41)	31 (55)	4 (7)	20 (36)	1 (2)	0
Georgia	Jan. 2013–Dec. 2017	201	127 (63)	74 (37)	74 (37)	8 (4)	76 (38)	0	5 (2)
Ukraine	Dec. 2019–Oct. 2020	191	113 (59)	78 (41)	78 (41)	28 (11)	38 (20)	2 (1)	1 (1)
Total	Jan. 2013–Oct. 2020	812	505 (62)	307 (38)	497 (61)	88 (11)	212 (26)	4 (<1)	11 (1)

Period: Time frame in which the samples were collected. Patients: Number of patients included and percentage within brackets. Other: Nosocomial infection (1 case, Moscow), mother-to-child transmission (1 case, Krasnodar), or blood products (2 cases, Ukraine). Abbreviations: Mos: Moscow; Krd: Krasnodar; HTX: heterosexual; MSM: men who have sex with men; IDU: intravenous drug users; Unkn: unknown.

**Table 2 viruses-14-02099-t002:** Distribution of HIV-1 subtypes, sub-subtypes, and recombinant forms in the different countries and regions studied.

	A6	B	CRF02_AG	CRF63_02A6	URF	Other	Total
Russia (Mos)	288 (79.1)	41 (11.3)	16 (4.4)	1 (1.6)	5 (1.4)	8 (2.2)	364
Russia (Krd)	43 (76.8)	6 (10.6)	0	2 (3.6)	5 (8.9)	0	56
Georgia	198 (98.5)	1 (0.5)	0	0	2 (1.0)	0	201
Ukraine	177 (92.7)	11 (5.8)	0	0	2 (1.0)	1 (0.5)	191
Total	706 (86.9)	59 (8.4)	16 (2.0)	8 (1.0)	14 (2.0)	9 (1.1)	812

Number of patients included are presented and percentage within brackets. Mos: Moscow; Krd: Krasnodar; CRF: circulating recombinant form; URF: unique recombinant form. Other: Genotype D (1 case), CRF03_AB (2 cases), and G (6 cases).

**Table 3 viruses-14-02099-t003:** Distribution of HIV-1 subtypes among the transmission categories.

	A6	B	CRF02_AG	CRF63_02A6	URF	Other
HTX	446 (90)	29 (6)	5 (1)	6 (1)	8 (2)	4 (1)
MSM	57 (65)	21 (24)	3 (3)	0	4 (5)	3 (3)
IDU	191 (90)	7 (3)	8 (4)	2 (1)	2 (1)	2 (1)
Other	2 (67)	1 (33)	0	0	0	0
Unknown	10 (91)	1 (9)	0	0	0	0

Number of patients are indicated. Numbers in brackets: Percentage. Other (subtype): subtype D (1 case), CRF03_AB (2 cases), subtype G (6 cases). Transmission categories: HTX: heterosexual; MSM: men who have sex with men; IDU: intravenous drug users; Other: Nosocomial infection (1 case), mother-to-child transmission (1 case), or blood products (2 cases). URF: unique recombinant form.

## Data Availability

All sequences used in this study are available through Genbank via accession numbers: OK474695-OK474699; OK474701-OK474704; OK474706-OK474707; OK474710-OK474712; OK474714-OK474721; OK474723-OK474735; OK474738-OK474757 (Krasnodar sequences). MW756383-MW756390, MW756393-MW756414, MW756416-MW756419, MW756421-MW756427, OL792300-OL792570, OL792574-OL792612 (Moscow sequences), OP352923-OP353328 (Ukrainian and Georgian *Protease* and *reverse transcriptase* sequences), and OP353329-OP353518 (Ukrainian and Georgian *Integrase* sequences). Data that were not made publicly available due to privacy issues related to pseudonymisation are available on request from the corresponding author.

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
