# Peer review of "Molecular Epidemiology of HIV-1 in Eastern Europe and Russia"

_viruses, 2022, doi:10.3390/v14102099_

Round 1
Reviewer 1 Report
In this study Maarten A.A. van de Klundert et al. provides updated insights into the molecular epidemiology of HIV-1 in Eastern Europe and Russia. Despite the fact that the rates and overall numbers of people diagnosed with HIV is the highest in the eastern part of Europe, limited information is available about this topic in literature. In spite of the small number of studied samples, the changing pattern of transmission routes between different risk groups and other findings about HIV subtypes and drug resistance supports the epidemiological relevance of this study.
The manuscript is mostly clearly written; however, there is some missing information and some points need to be revised to improve the study.
General concept comments:
- In the Introduction I suggest giving some information about the practice of initial ART in the listed countries (“WHO Treat All”, most common initial combinations, backbone of ART, etc.). How relevant is the application of mentioned CAB+NNRTI in initial therapy?
- Please indicate the clinical relevance of drug resistance mutations in section 3.5.
- The results of the drug resistance analysis should be discussed in more detail (comparison of the results with surrounding countries/regions, and/or with earlier studies from same countries to follow up the changes of patterns of drug resistance; e.g. Kirichenko et al, PLoS ONE, 2022;17(1):e0257731).
Specific comments:
Abstract:
- I suggest using the term „heterosexual contact” instead of „heterosexual” (line 38), and „strand transfer inhibitor” instead of „strand inhibitor” (line 42).
- Because of the limited number of studied patients and self-reporting of possible route of infection in this study, the conclusions in the Abstract should be softened (for example „exchange of viruses among different risk groups can be observed/ is possible/ may occur”).
Introduction section:
- According to the report entitled HIV/AIDS surveillance in Europe 2021 the rate of new infections in Western Europe is 3.7 per 100 000 population. Please correct the infection rate in line 55. If this data does not concern to Western Europe, then please indicate the geographical region. Insertion of the term “per 100 000” after 32.6 (line 57) is recommended.
Method section:
- Please give the relevant reference of in house method mentioned in line 121, or describe it shortly in the manuscript.
- The reference of Bioedit (line 124-126) should be listed in square brackets and with the relevant number.
- If you used Bootstrap method for phylogeny test during constructing neighbour-joining tree, please indicate the number of Bootstrap replicates.
- Statistical methods used in GraphPad Prism and the threshold of statistical significance should be listed.
- It is not clear, whether the authors used HIV-BLAST (https://www.hiv.lanl.gov/content/sequence/BASIC_BLAST/basic_blast.html) to determine the subtype of studied sequences, or used NCBI BLAST, and then another method to determine the subtype of most similar sequences (line 161-164).
- The first sentence of recombination analysis should be rephrased (line 173-175).
Results section:
- As authors describe age as median, they should indicate the interquartile range, and not the range.
- If authors used statistical significance test to examine the proportion of MSM population in Georgia and Ukraine (line 192-194), then please indicate the p-value.
- The sentence “The heterosexually infected dominated in all three countries.” (line 191-192) should be deleted, since the meaning is the same with the sentence “In all groups, the dominant transmission route was heterosexual contact (61.3%).” (line 189-190).
- The aggregation of inclusion periods of different countries is not recommended in Table 1.
- Comparing the proportion of different risk groups between studied persons and all newly diagnosed persons in the countries is recommended to corroborate the representativeness of the study.
- Please mark the clusters of different subtypes in Figure 2A (as listed in the legend).
- Figure caption for 2C is missing.
- In Table S3 a legend explaining the highlights (green background, asterisk) is missing.
- In the first paragraph of section 3.6. A64V mutation is analyzed, while the related Figure S3A contains sequences with A62V mutation.
Discussion section:
- Authors refer to earlier studies and data in line 366, 380, 401, but no reference is given, please list the missing publications.
Author Response
Point-by-point rebuttal reviewer 1:
Dear reviewer,
Many thanks for your kind words and the points that you raised to improve our manuscript. Please find our responses indicated in red in this document.
In this study Maarten A.A. van de Klundert et al. provides updated insights into the molecular epidemiology of HIV-1 in Eastern Europe and Russia. Despite the fact that the rates and overall numbers of people diagnosed with HIV is the highest in the eastern part of Europe, limited information is available about this topic in literature. In spite of the small number of studied samples, the changing pattern of transmission routes between different risk groups and other findings about HIV subtypes and drug resistance supports the epidemiological relevance of this study.
The manuscript is mostly clearly written; however, there is some missing information and some points need to be revised to improve the study.
General concept comments:
- In the Introduction I suggest giving some information about the practice of initial ART in the listed countries (“WHO Treat All”, most common initial combinations, backbone of ART, etc.). How relevant is the application of mentioned CAB+NNRTI in initial therapy?
We have added the following to the introduction: recommendations to physicians, the preferred and most used initial combinations, as well as the notion that CAB is not currently used as 1st regimen.
- Please indicate the clinical relevance of drug resistance mutations in section 3.5.
We included the background information on the DRMs in section 3.5.
- The results of the drug resistance analysis should be discussed in more detail (comparison of the results with surrounding countries/regions, and/or with earlier studies from same countries to follow up the changes of patterns of drug resistance; e.g. Kirichenko et al, PLoS ONE, 2022;17(1):e0257731).
We have extended the discussion with references (amongst others the one mentioned) and discussed our findings in the context of these previous findings. Briefly, we found higher prevalence of pDRMs, although due to the relatively small sample size of our study as well as the other studies, we refrain from drawing too strong conclusions.
Specific comments:
Abstract:
- I suggest using the term „heterosexual contact” instead of „heterosexual” (line 38), and „strand transfer inhibitor” instead of „strand inhibitor” (line 42).
We agree and have adapted the text accordingly.
- Because of the limited number of studied patients and self-reporting of possible route of infection in this study, the conclusions in the Abstract should be softened (for example „exchange of viruses among different risk groups can be observed/ is possible/ may occur”).
We have changed the text to “may occur more often than earlier reported”.
Introduction section:
- According to the report entitled HIV/AIDS surveillance in Europe 2021 the rate of new infections in Western Europe is 3.7 per 100 000 population. Please correct the infection rate in line 55.
Many thanks for pointing out this mistake, we have corrected it.
If this data does not concern to Western Europe, then please indicate the geographical region. Insertion of the term “per 100 000” after 32.6 (line 57) is recommended.
We have followed your recommendation and adapted the text accordingly.
Method section:
- Please give the relevant reference of in house method mentioned in line 121, or describe it shortly in the manuscript.
The reference has been added.
- The reference of Bioedit (line 124-126) should be listed in square brackets and with the relevant number.
Indeed, we adapted this.
- If you used Bootstrap method for phylogeny test during constructing neighbour-joining tree, please indicate the number of Bootstrap replicates.
No phylogeny test was performed, so no number of bootstraps was set. Instead, the neighbour-joining tree was merely used to test whether the topology of the tree generated with the nextstrain software was correct. The relevant paragraph in the materials and methods section was adapted to incorporate this information.
- Statistical methods used in GraphPad Prism and the threshold of statistical significance should be listed.
In the end, only numerical analysis was performed, so the reference to statistical analysis has been removed from the manuscript and the text has been adapted to reflect this notion and correctly describe how Excel and GraphPad were used.
- It is not clear, whether the authors used HIV-BLAST (https://www.hiv.lanl.gov/content/sequence/BASIC_BLAST/basic_blast.html) to determine the subtype of studied sequences, or used NCBI BLAST, and then another method to determine the subtype of most similar sequences (line 161-164).
HIV-BLAST was used, we have adapted the text to make this clear (by using “HIV-BLAST” instead of just “BLAST”) and have adapted the reference to refer to the hiv.lanl.gov website rather than to the general NCBI BLAST website to stress this notion.
- The first sentence of recombination analysis should be rephrased (line 173-175).
This sentence was indeed unclear and has been rephrased.
Results section:
- As authors describe age as median, they should indicate the interquartile range, and not the range.
We added the interquartile range.
- If authors used statistical significance test to examine the proportion of MSM population in Georgia and Ukraine (line 192-194), then please indicate the p-value.
We decided to not perform statistical analysis and have deleted the notion about the proportion of MSM in Ukraine. The lines have been adapted accordingly.
- The sentence “The heterosexually infected dominated in all three countries.” (line 191-192) should be deleted, since the meaning is the same with the sentence “In all groups, the dominant transmission route was heterosexual contact (61.3%).” (line 189-190).
True, the sentence has been deleted.
- The aggregation of inclusion periods of different countries is not recommended in Table 1.
It is important to point out that the inclusion for Georgia differed from the other inclusions, the other reviewer stressed this notion so we would propose to leave it in.
- Comparing the proportion of different risk groups between studied persons and all newly diagnosed persons in the countries is recommended to corroborate the representativeness of the study.
Comparable (i.e., self-reported) numbers are available for Russia, covering all infections up to 2022. These numbers and a reference to the report that contains them have been added to the manuscript results section, and a paragraph describing the comparison has been added to the discussion. Briefly, we find that IDU is reported less, and MSM is reported more in our study. This may reflect differences in what is considered socially acceptable rather than how people are infected, but this is beyond the scope of our study.
- Please mark the clusters of different subtypes in Figure 2A (as listed in the legend).
The figure legend was adapted to more accurately describe the clusters, for the sake of simplicity we would rather not include these in the figure itself.
- Figure caption for 2C is missing.
This has been added.
- In Table S3 a legend explaining the highlights (green background, asterisk) is missing.
The green highlights were removed (had to do with prior analysis) and the asterisk has been explained in the revised table S3. The asterisk referred to a patient with multiple drug resistance mutations. This patient has been mentioned in the body text and a reference to table S3 has been added.
- In the first paragraph of section 3.6. A64V mutation is analysed, while the related Figure S3A contains sequences with A62V mutation.
This was a typo in the text, which has been corrected.
Discussion section:
- Authors refer to earlier studies and data in line 366, 380, 401, but no reference is given, please list the missing publications.
The requested references have been added.

Reviewer 2 Report
With the new FDA-approved CAB+RPV drug regimen available, studies of the prevalence and transmission of the of-concern A6 subpopulation and drug resistance is of tremendous importance to the health care and research communities. Maarten et al. attempt to fill this gap and present the results in an intuitive way; however, I have some concerns with the methodology that should be addressed before moving forward.
The tree reconstruction method is not entirely clear. The authors state that the initial tree was generated using IQ-TREE, only after stating that a NJ tree was reconstructed in MEGA. Also, there was no confirmation that there was enough phylogenetic signal (e.g., using likelihood mapping) or temporal signal (e.g., using root-to-tip regression) prior to tree dating. Is this perhaps why outgroup rooting was used? If sampling dates are known, highly divergent sequences (i.e., entirely different clade reference sequences) should be avoided, as they can bias clock calibration. It is mentioned that this rooting strategy had no major impact on tree topology, but what criterion was used here, and did it impact internal node time estimates? There is a reference to Schlosser et al (ref 18) from the same group meant to describe this in more detail, but neither the approach nor rationale for this approach is mentioned in this reference.
The authors note that the “largest transmission route was from Ukraine to Moscow” (line 292), insinuating that routes involving Georgia were less prevalent; however, the authors need to be careful here because Georgia was sampled during a time period (2013-2017) that was distinct from the similar time frame of sampling for Ukraine and Russia (2019-2020). Hence, missing samples from Georgia during 2019-2020 might actually change this story significantly. Also, how are the authors describing directionality with “transmissions” (e.g., “Despite many transmissions from Ukraine to Krasnodar and Georgia, very few transmissions from these regions to Ukraine took place”, line 294)? The major concerns here are as follows: 1) This directionality is not pictured (nor the statistical support) in Fig 2 and 2) As these transmissions are inferred from internal branches within the tree, descriptors such as "inferred", “unobserved” or “unsampled” should be used here to distinguish ancestral state reconstruction from classical epidemiological investigation of transmission events. And, again, quantification of these inferred transmission events should be discussed with limitations, since the dataset is biased towards Russia and, again, the sampling time with regards to these locations is not uniform across locations.
The discussion will also needs to be reworded to take these limitations into account.
Author Response
Point-by-point rebuttal reviewer 2:
Dear reviewer,
Many thanks for your kind words and the points that you raised to improve our manuscript. Please find our responses indicated in red in this document.
With the new FDA-approved CAB+RPV drug regimen available, studies of the prevalence and transmission of the of-concern A6 subpopulation and drug resistance is of tremendous importance to the health care and research communities. Maarten et al. attempt to fill this gap and present the results in an intuitive way; however, I have some concerns with the methodology that should be addressed before moving forward.
The tree reconstruction method is not entirely clear. The authors state that the initial tree was generated using IQ-TREE, only after stating that a NJ tree was reconstructed in MEGA.
Regarding the tree reconstruction, all trees presented in the manuscript were generated using IQ-TREE in the nextstrain workflow. Since we didn’t work/haven’t published work based on analysis in Nextstrain, we felt we should verify the outcome using a method that we were more familiar with (MEGA). We compared the trees at essential nodes and some random tips, only to conclude correct running of the nextstrain workflow. We adapted the text concerned to better clarify this notion.
Also, there was no confirmation that there was enough phylogenetic signal (e.g., using likelihood mapping) or temporal signal (e.g., using root-to-tip regression) prior to tree dating. Is this perhaps why outgroup rooting was used? If sampling dates are known, highly divergent sequences (i.e., entirely different clade reference sequences) should be avoided, as they can bias clock calibration.
We did not date any of the trees in the manuscript, as like you say there are highly divergent sequences in the dataset. Also, the non-uniform mixing because the samples are collected in different areas would complicate such analysis. Therefore, we represent the data with the branch lengths as divergence rather than time. To further clarify this, we adapted the manuscript text and included this notion in the materials and methods section and in the figure legends.
It is mentioned that this rooting strategy had no major impact on tree topology, but what criterion was used here, and did it impact internal node time estimates?
We only looked at the tree topology, i.e., the localisation and content of clusters. The internal node time estimates may have been affected but were not analysed, as we have made clear in the revised text.
There is a reference to Schlosser et al (ref 18) from the same group meant to describe this in more detail, but neither the approach nor rationale for this approach is mentioned in this reference.
This is correct, the reference has been removed.
The authors note that the “largest transmission route was from Ukraine to Moscow” (line 292), insinuating that routes involving Georgia were less prevalent; however, the authors need to be careful here because Georgia was sampled during a time period (2013-2017) that was distinct from the similar time frame of sampling for Ukraine and Russia (2019-2020). Hence, missing samples from Georgia during 2019-2020 might actually change this story significantly.
This is correct, we have added this notion to the text.
Also, how are the authors describing directionality with “transmissions” (e.g., “Despite many transmissions from Ukraine to Krasnodar and Georgia, very few transmissions from these regions to Ukraine took place”, line 294)? The major concerns here are as follows: 1) This directionality is not pictured (nor the statistical support) in Fig 2
The directionality is pictured by the colour of the connecting lines, we added this notion to the figure legend.
and 2) As these transmissions are inferred from internal branches within the tree, descriptors such as "inferred", “unobserved” or “unsampled” should be used here to distinguish ancestral state reconstruction from classical epidemiological investigation of transmission events. And, again, quantification of these inferred transmission events should be discussed with limitations, since the dataset is biased towards Russia and, again, the sampling time with regards to these locations is not uniform across locations.
We have moved the precise description of how the transmission routes were inferred from the figure legend to the main text, and we have explicitly pointed out the limitations you brought forward.
The discussion will also needs to be reworded to take these limitations into account.
The limitations have now been explicitly added to the discussion.

Round 2
Reviewer 2 Report
We did not date any of the trees in the manuscript, as like you say there are highly divergent sequences in the dataset. Also, the non-uniform mixing because the samples are collected in different areas would complicate such analysis. Therefore, we represent the data with the branch lengths as divergence rather than time. To further clarify this, we adapted the manuscript text and included this notion in the materials and methods section and in the figure legends.
I think the authors missed my point here - outgroup rooting can be more risky than clock rooting, as it can potentially introduce long branch attraction. If clock rooting was not used, despite the fact that there are dates available, then the authors must state more explicitly why they did not do so. The authors have revised their text to say that the sequences were "too divergent," but it is unclear what is meant by this, as there are no figures or data to support this. Were the authors meaning that the variance for the root-to-tip regression was high? If so, relaxed clocks are very robust to moderately high deviation.
I would also highly recommend a more statistical approach to the phylogeography, that takes into account multiple possible tree topologies for the data (BEAST), especially since the data are not very large. However, since the results can be readily explained, this is not a concern for publication.
